# Patient-Level Classification from Volumetric CT: A Comparative Study of Foundation Models and 3D CNNs

**Sarmad Ahmad Khan**[1*]                SarmadAhmad.Khan@muehlenkreiskliniken.de
**Ph.D. Iram Shahzadi**[1*]                    Iram.Shahzadi@muehlenkreiskliniken.de
**Dr. med. Julius Henning Niehoff**[1]        Julius.Niehoff@muehlenkreiskliniken.de
**Dr. med. Jan Robert Kroeger**[1]      JanRobert.Kroeger@muehlenkreiskliniken.de
**Prof. Dr. med. Alexey Surov**[1]              Alexey.Surov@muehlenkreiskliniken.de
**Prof. Dr. med. Jan Borggrefe**[1]          Jan.Borggrefe@muehlenkreiskliniken.de

[1] *Department of Radiology, Neuroradiology and Nuclear Medicine, Johannes Wesling University Hospital, Ruhr University Bochum, 44801 Bochum, Germany*

## Abstract

Foundation models may improve medical image classification by providing transferable visual representations in limited-data settings. We evaluate MedSigLIP for patient-level classification for normal condition and four conditions of abdominal pain including appendicitis, cholecystitis, sigmoid diverticulitis and free space (perforation) from volumetric CT and compare it with a MedicalNet-pretrained 3D ResNet-18 baseline. Using attention-based pooling for slice aggregation, we study frozen linear probing and LoRA-based adaptation across dataset fractions from 20% to 100%. MedicalNet achieves the best full-data performance (macro F1 0.527) with full fine-tuning, whereas linear probing in MedSigLip is the most stable across data regimes (0.464–0.511). LoRA increases computational cost without consistent gains. These findings emphasise the current limitations of off-the-shelf vision-language foundation models for volumetric CT classification, and highlight the continued importance of domain-specific 3D pre-training, effective slice aggregation and adaptation strategies tailored to medical imaging.

**Keywords:** foundation models, volumetric CT, patient-level classification, MedSigLIP, MedicalNet, LoRA, attention pooling

## 1. Introduction

The relevance of deep learning to abdominal CT is increasing, particularly in cases of acute abdominal pain, where CT is the primary imaging modality for rapid evaluation of diverse urgent pathologies. In this setting, CT is commonly utilised due to its capacity to facilitate comprehensive evaluation of diverse clinical conditions, including appendicitis, cholecystitis, and perforation. Three-dimensional convolutional neural networks are well suited to volumetric CT because they capture cross-slice spatial context. However, they typically require substantial annotated data and computational resources (Litjens et al., 2017; Çiçek et al., 2016; Chen et al., 2019; Wolfe et al., 2022).

We aim to assess whether foundation-model representations can provide a practical alternative in this setting. In computer vision, these have shown excellent results across tasks (Radford et al., 2021), and similar approaches are increasingly being used in biomedical imaging (Zhai et al., 2023). Hence, we investigate MedSigLIP for patient-level multi-class classification from abdominal CT and compared frozen linear probing and LoRA-based

adaptation of MedSigLIP against a fully fine-tuned MedicalNet-pretrained 3D ResNet-18 baseline (Chen et al., 2019; Hu et al., 2021).

Considering that diagnostically relevant findings may be subtle and distributed across multiple slices, we use attention-based pooling for patient-level aggregation following the multiple instance learning framework of Ilse et al. (2018). To avoid introducing anatomical bias through explicit region-of-interest selection, the primary analysis is performed on full patient volumes. Entropy-based slice selection is treated as a secondary exploratory analysis and is reported in the appendix.

## 2. Methods

We study a five-class patient-level CT classification task comprising appendicitis (237), cholecystitis (185), sigmoid diverticulitis (428), free air (perforation) (59), and normal (450) cases. To evaluate data efficiency, experiments are performed on dataset fractions of 20%, 40%, 60%, 80%, and 100% of the available patients, using class-aware sampling and patient-level train-validation-test partitioning. This design enables systematic comparison of the methods across varying training set sizes while preserving class representation.

For MedSigLIP-based experiments, each patient volume is represented in a 2.5D slice-based format in which consecutive axial slices are combined to preserve limited inter-slice context. Each slice is independently encoded by the MedSigLIP vision backbone, producing a sequence of slice-level embeddings. These embeddings are then aggregated using an attention-based pooling module, which assigns a weight to each slice and forms a weighted patient-level representation (Ilse et al., 2018). This pooled representation is subsequently classified using a linear prediction head trained with class-weighted cross-entropy loss. Two MedSigLIP transfer settings are considered: linear probing and LoRA.

In the linear probing configuration, the pretrained MedSigLIP encoder remains frozen and only the attention-pooling and classifier layers are optimized. In the LoRA configuration, low-rank trainable adapters are introduced into selected encoder layers to enable parameter-efficient adaptation of the pretrained model (Hu et al., 2021).

As a volumetric baseline, we employ a 3D ResNet-18 pretrained on MedicalNet (Chen et al., 2019) which goes classification head adaptation followed by full fine-tuning. In contrast to the slice-based MedSigLIP framework, this model operates directly on volumetric inputs and learns cross-slice structure through three-dimensional convolution. Training follows a two-stage strategy with classifier warm-up and subsequent fine-tuning. Across all models, performance is assessed using macro F1-score as well as balanced accuracy which is crucial to justify the proper representation of all the classes based on the average of sensitivity and specificity, and model selection is based on validation performance. Computational comparison is considered in terms of training efficiency, parameter adaptation, and practical resource requirements.

## 3. Results and Discussion

**Linear probing** with full slices yields the most stable performance across dataset fractions, achieving macro F1-scores of 0.464 at 20% data and 0.511 at 100%, with intermediate values ranging from 0.419 to 0.486. This stability demonstrates robust frozen MedSigLIP repre-

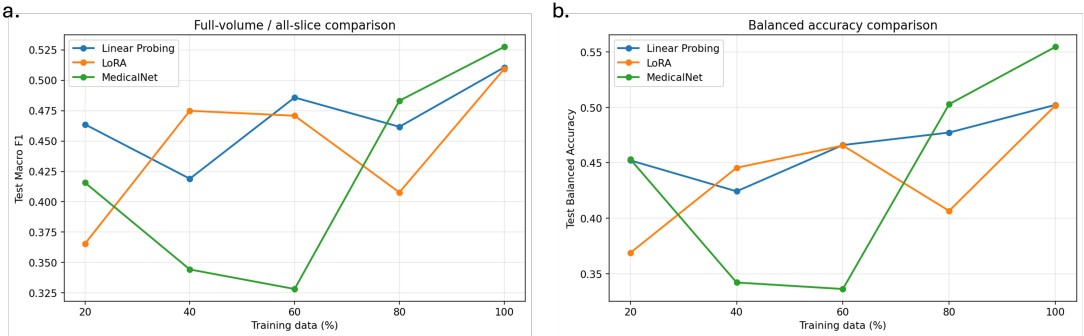

Figure 1: Performance comparison for linear probing MedSigLip, LoRA-adapted Med-SigLip, and MedicalNet3D. (a) Macro F1-score. (b) Balanced accuracy.

sentations for patient-level classification without encoder adaptation. Entropy-based slice selection (appendix) follows similar trends but underperforms full-slice probing, particularly with smaller subsets, indicating that reduced slice diversity discards diagnostically relevant volumetric information.

The **MedicalNet 3D** baseline is computationally efficient and attains the highest full-data macro F1-score (0.527). However, it exhibits dataset-size dependence, with lower scores at 40% (0.344) and 60% (0.328), then recovery at 80% (0.484). This confirms that task-specific volumetric pretraining remains competitive when sufficient training data are available (Chen et al., 2019).

**LoRA fine-tuning** increases computational cost without consistent benefit. In the full-slice setting, LoRA yields macro F1-scores of 0.365 (20%), 0.468 (40%), 0.475 (60%), 0.410 (80%), and 0.511 (100%). Although matching linear probing at full data, its intermediate instability fails to justify the additional resource requirements. LoRA's stronger performance under entropy-based selection (appendix) suggests sensitivity to reduced input diversity during adaptation. Attention-pooling effectiveness influences MedSigLIP results: suboptimal weighting may emphasize non-informative slices, limiting performance gains over MedicalNet at higher data volumes. Qualitative attention analysis is provided in appendix.

**Conclusion:** We systematically compared foundation-model transfer learning and volumetric pretraining for patient-level CT classification. Overall, linear probing offers the best trade-off between performance (0.464–0.511 macro F1), stability, and computational cost. MedicalNet achieves peak full-data performance (0.527) but requires larger datasets. LoRA incurs highest overhead without reliable gains. Model selection should therefore consider dataset size and resource constraints.

**Future work:** We will quantify pretraining contributions by including non-pretrained baselines, explore additional architectures (e.g., DenseNet-based models) and alternative foundation models, investigate unbiased slice-selection strategies for clinical applicability, and examine alternative aggregation mechanisms and cross-modal representations to further understand global versus local feature learning in volumetric data.

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

## Appendix

### Appendix A. Entropy-based Slice Selection

To reduce computational cost, we explored entropy-based slice selection as a secondary experiment, using slice-wise Shannon entropy as a heuristic measure of image information content (Schielein et al., 2016). For a slice with normalized intensity histogram probabilities $p_i$, entropy is defined as

$$H = -\sum_{i=1}^{N} p_i \log(p_i), \tag{1}$$

where $N$ denotes the number of histogram bins. In implementation, a small constant may be added inside the logarithm for numerical stability.

For linear probing, entropy selection with $k = 24$ yielded macro F1-scores ranging between 0.390 and 0.438 across 20% to 100% data, while $k = 64$ yielded between 0.241 and 0.445. Both settings underperformed relative to full-slice linear probing, although $k = 64$ partially recovered performance at higher data fractions.

For LoRA, entropy-based slice selection produced macro F1-scores ranging between 0.353 and 0.438 for $k = 24$, with an exception at 40% where it rises to 0.527, and between 0.338 and 0.500 for $k = 64$, with a similarly strong value of 0.502 at 40%. This jump at 40% may reflect improved attention-pooling behavior under a reduced-noise slice subset. These results suggest that larger slice subsets are more favorable for LoRA and that aggressive slice reduction may remove relevant diagnostic context needed for stable adaptation, but if carefully chosen can potentially yield better results. Overall, entropy-based slice selection is useful as an exploratory strategy for reducing resource consumption, but it does not alter the main conclusions of the study.

Table 1: Patient-level classification performance (macro F1) across models and dataset fractions.

| Setting | 20% | 40% | 60% | 80% | 100% |
|---|---|---|---|---|---|
| Linear Probing — All slices | **0.46** | 0.42 | 0.48 | 0.46 | 0.51 |
| Linear Probing — Entropy $k = 24$ | 0.39 | 0.39 | 0.44 | 0.42 | 0.36 |
| Linear Probing — Entropy $k = 64$ | 0.24 | 0.36 | 0.37 | 0.46 | 0.44 |
| LoRA — All slices | 0.36 | 0.47 | 0.47 | 0.41 | 0.51 |
| LoRA — Entropy $k = 24$ | 0.35 | **0.53** | 0.42 | 0.42 | 0.44 |
| LoRA — Entropy $k = 64$ | 0.34 | 0.50 | **0.49** | **0.50** | 0.46 |
| MedicalNet — 3D volume | 0.42 | 0.34 | 0.33 | 0.48 | **0.53** |

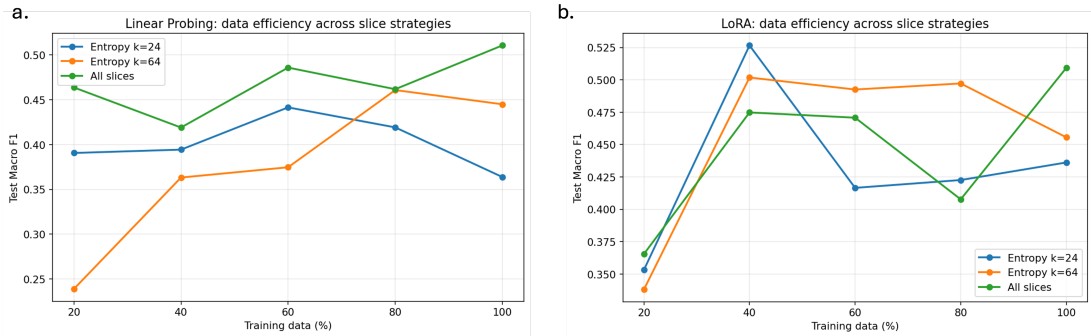

Figure 2: Performance comparison for different slice-selection configurations in (a) linear probing MedSigLip and (b) LoRA-adapted MedSigLip.

## Appendix B. Attention Comparison

Because attention-based pooling is the core aggregation mechanism in the MedSigLIP framework, we further compare attention-weight distributions with predicted class probabilities and ground-truth labels. This comparison is provided in the supplementary figure to illustrate cases in which suboptimal slice weighting may contribute to reduced performance of MedSigLIP-based approaches relative to MedicalNet.

To aggregate slice-level representations into a patient-level prediction, attention pooling computes a weighted sum of slice embeddings:

$$\mathbf{z}_{\text{patient}} = \sum_{i=1}^{N} \alpha_i \mathbf{z}_i, \tag{2}$$

where $\mathbf{z}_i$ denotes the embedding of slice $i$ and $\alpha_i$ is its attention weight. The attention weights are obtained by

$$\alpha_i = \frac{\exp\big(\mathbf{w}^\top \tanh(\mathbf{V}\mathbf{z}_i)\big)}{\sum_{j=1}^{N} \exp(\mathbf{w}^\top \tanh(\mathbf{V}\mathbf{z}_j))}, \tag{3}$$

where $\mathbf{V}$ is a learnable projection matrix and $\mathbf{w}$ is a learnable attention vector. This formulation allows the model to assign higher importance to diagnostically relevant slices when constructing the final patient representation.

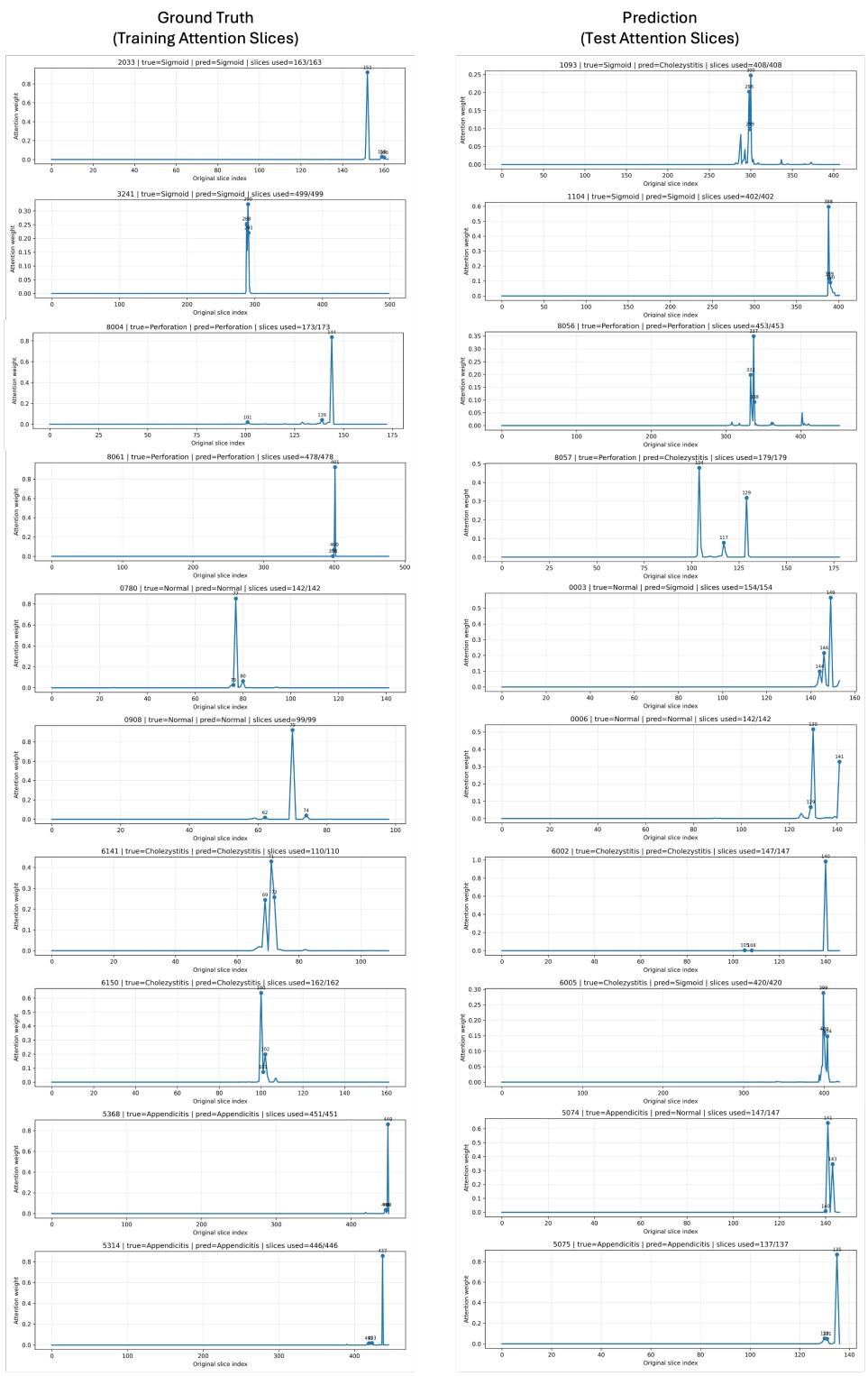

Figure 3: Comparison of attention pooling over effective slices during decision-making by linear probing MedSigLip. (a) Ground truth. (b) Prediction examples.

