# OpenReview forum: "Patient-Level Classification from Volumetric CT: A Comparative Study of Foundation Models and 3D CNNs"
_MIDL.io/2026/Short_Papers — MIDL 2026 - Short Papers Poster_

### Official Review · Reviewer_FNtu · 2026-05-04
**Comparison study between fine-tuned FMs and in-domain models**

**Rating:** 4
**Confidence:** 4

**Review:**

see below

**Summary:**

This paper conducts a comparative study between a classification model trained and tested on the same domain, and a generic medical foundation model fine-tuned (either using linear probing or LoRA) on this domain.
Overall, the fine-tuned foundation models prove to be more stable in low-data regime, while at full dataset availability, the specific model is more accurate.

**Strengths:**

1. interesting and timely study to compare expert models against fine-tuned foundation models.
2. the conclusions are interesting and bring insight to a problem that is affecting a large part of our community.

**Weaknesses:**

1. each network should be trained several times to average out bad initialisation. This could explain the huge drops in performance of some of the presented models.

2. This study is interesting but the experimental setup is too narrow (only 1 task, 1 dataset, 1 foundation model) to draw more general conclusions.

3. I think future work should conduct the same comparison on out-of-domain datasets, to verify potential claims of better generalisation of fine-tuned foundation models.

4. linear probing is generally used to describe classification results only using 1 linear layer, not a whole classifier.

**Justification Of Rating:**

This paper will be of high interest to the MIDL community, and I recommend acceptance despite the few experimental shortcomings listed above.

---

### Decision · Program_Chairs · 2026-05-08

Accept (Poster)